

# A destructive active defense algorithm for deepfake face images

Yang Yang[1], Norisma Binti Idris[1], Chang Liu[2], Hui Wu[2] and Dingguo Yu[3]

[1] Faculty of Computer Science and Information Technology, Universiti Malaya, Kuala Lumpur, Malaysia
[2] Institute of Intelligent Media Technology, Communication University of Zhejiang, Hangzhou, China
[3] College of Media Engineering, Communication University of Zhejiang, Hangzhou, China

## ABSTRACT

The harm caused by deepfake face images is increasing. To proactively defend against this threat, this paper innovatively proposes a destructive active defense algorithm for deepfake face images (DADFI). This algorithm adds slight perturbations to the original face images to generate adversarial samples. These perturbations are imperceptible to the human eye but cause significant distortions in the outputs of mainstream deepfake models. Firstly, the algorithm generates adversarial samples that maintain high visual fidelity and authenticity. Secondly, in a black-box scenario, the adversarial samples are used to attack deepfake models to enhance their offensive capabilities. Finally, destructive attack experiments were conducted on the mainstream face datasets CASIA-FaceV5 and CelebA. The results demonstrate that the proposed DADFI algorithm not only improves the generation speed of adversarial samples but also increases the success rate of active defense. This achievement can effectively reduce the harm caused by deepfake face images.

## BACKGROUND

Deepfake is a method that uses deep learning technologies such as generative adversarial networks (*Brophy et al., 2023*) to create synthetic digital content, including images. Fake news (*Phan, Nguyen & Hwang, 2023*) generated by deepfake can mislead public information judgment and decision-making. Additionally, deepfake-created fake pornographic content (*Sha et al., 2023*) can infringe on the rights of the individuals depicted, while fake evidence (*Shukla & Goh, 2024*) produced by deepfake can undermine legal justice. Due to the growing concerns about information authenticity, artificial intelligence ethics, and social trust raised by deepfake technology, an increasing number of scholars are focusing on how to balance technological development with the protection of personal rights.

Given the above challenges and the increasingly complex deepfake problem, researchers are diligently developing deepfake detectors to address these issues and enhance detection robustness. The goal is to identify face images created by continuously evolving high-quality deepfake digital content. Among the current mainstream research on deepfake detection, the more mature approaches are based on passive detection methods. Examples include

Corresponding author
Dingguo Yu, yann@cuz.edu.cn

the method based on physiological signals proposed by *D'Amelio et al. (2023)*, the method based on traditional image forensics proposed by *Zhao et al. (2023)*, and the method based on image tampering traces proposed by *Wu, Liao & Ou (2023)*. A common perspective among these methods is that they utilize characteristic information extracted directly from the deepfake digital content itself to achieve authenticity identification.

Compared with the above methods, active defense can theoretically detect and mitigate deepfake problems earlier by embedding hidden information such as adversarial samples and utilizing strategies like watermark traceability. Additionally, active defense employs adversarial attack methods to generate offensive adversarial samples, which are visually indistinguishable to humans but cause significant distortion in the digital content output by deepfake models. For instance, *Yuan et al. (2024)* proposed a semi-fragile digital watermark generation method based on deep learning. Semi-fragile watermarks are sensitive to operations like image compression and rotation, and are particularly responsive to image modifications, allowing easy authenticity verification of digital content. *Sun et al. (2023b)* demonstrated the transferability of digital watermarks between training data and deepfake models, achieving deepfake traceability by embedding digital watermarks in the target digital content. *Radanliev & Santos (2023)* utilized the Jacobian matrix to identify key pixels in adversarial attacks, exploring adversarial samples with superior attack performance. *Dimlioglu & Choromanska (2024)* combined the gradient descent heuristic algorithm to enhance the generalization of adversarial samples by uniformly weighting the gradient across regions and models. *Ouyang et al. (2023)* proposed a cross-model universal attack watermark generation method, addressing conflicts between watermarks generated by different models through the design of attack channels and two-level fusion strategies.

# INTRODUCTION

Due to the substantial differences among various deepfake models, the aforementioned attack algorithms achieve optimal results primarily in black-box scenarios. Additionally, the adversarial examples generated by these algorithms often lack sufficient transferability and require access to the deepfake model's output for iterative updates, resulting in relatively low output efficiency of the adversarial examples.

Given the limitations and challenges of existing deepfake active defense detection methods, this paper innovatively proposes a destructive active defense algorithm for deepfake face images (DADFI) to address the low efficiency and poor transferability of current solutions. The specific steps of this algorithm are as follows:

(1) Generate adversarial samples: By extracting multi-dimensional feature information from images using a generative adversarial network (GAN), adversarial samples are generated. The quality of these samples is enhanced through adversarial loss strategies.

(2) Optimize adversarial samples: The adversarial attack algorithm is applied to the deepfake model, conducting black-box attacks in simulated black-box scenarios to optimize the attack performance of the adversarial samples.

(3) Improve attack capabilities: Training with multiple deepfake models is performed to enhance the generalization ability of the adversarial samples, improving their resilience to cross-model attacks.

The innovation of this algorithm lies in its ability to achieve end-to-end generation of adversarial samples based on a GAN. Unlike conventional adversarial attack algorithms that require frequent access to the deepfake model during the generation process, this algorithm eliminates the need to access the original deepfake model once the model training is complete. This approach significantly enhances the efficiency and transferability of the adversarial samples, providing a robust solution to the deepfake threat.

The paper is structured as follows: the Introduction provides an overview of the deepfake problem and its increasing impact, the motivation for developing efficient and transferable active defense mechanisms, and introduces the proposed DADFI along with its key innovations. The Related Works section is divided into three parts: research on deepfake based on face images, research on adversarial attacks and deepfake, and research on deepfake based on generative adversarial network, offering a comprehensive review of existing literature in these areas. The Algorithm Design section details the strategies for generating and fine-tuning adversarial samples, including the algorithm for generating adversarial examples and the algorithm for fine-tuning against adversarial examples. The Experiments section describes the models, datasets, and parameters used, along with the measurement methods for experimental results. The paper concludes with the Conclusion section, summarizing the findings.

## RELATED WORKS

### Research on deepfake based on face images

Currently, research on deepfake based on face images is in a stage of rapid development. In terms of algorithms, researchers are working hard to improve the authenticity and indistinguishability of generated face images, including improving the structure of generative adversarial networks, introducing attention mechanisms, and using multi-scale and multi-view training data. In terms of detection, researchers are also exploring more effective image recognition methods, including using deep learning models to extract image features, and methods that combine artificial intelligence and machine learning. The mainstream deepfake method based on face images is to tamper with the facial features of the face. By exchanging faces in the target face image and the original face image, the purpose of modifying the character's identity has been developed from traditional 3D reconstruction technology to deepfake technology based on generative adversarial networks (*Li et al., 2023b*). By digitally modifying all or part of the features in the target face image to achieve the purpose of faking specific expressions, traditional graphics technology has developed into technology based on deep learning (*Ali et al., 2024*).

*Kuang et al. (2021)* proposed a novel face replacement method, using a generative adversarial network to seamlessly replace the face in the original image with a synthetic face. The synthetic face looks as natural as an ordinary face, but is different from the original face. The appearance is completely different. *Wang et al. (2023)* extended face replacement technology to style transfer. This study proposed a multi-image style transfer loss function to improve the realism of generated face images. *Huang et al. (2023)* proposed a novel implicit identity-driven framework for face swap detection. This strategy designs

an explicit identity contrast loss and an implicit identity exploration loss, which supervises the convolutional neural network (CNN) to embed the face image into the implicit identity space, improving the face replacement effect. *Han et al. (2023)* proposed a method based on multi-classification tasks to distinguish multiple types of homologous deepfake face images. This method relies on the new network framework of FCD-Net, facial prominence saliency algorithm and contour detail feature extraction algorithm. *Waseem et al. (2023)* explored existing methods for deepfake images and videos for face and expression replacement, outlining publicly available datasets for benchmarking. With the development of deepfake technology, deepfake face images have become more sophisticated and difficult to identify, which has also created conditions for the abuse of deepfake and the leakage of personal privacy. *Akhtar (2023)* reviewed four types of deepfakes, including face manipulation (identity swapping), face re-creation, attribute manipulation, and whole face synthesis. For each category, the generation methods of deepfakes or face manipulations and the detection methods of these manipulations are described in detail.

## Research on adversarial attacks and deepfake

Nowadays, research on adversarial attacks against deepfake is constantly developing. Mainstream research focuses on the interpretability, robustness and development of detection algorithms of generative models, and has made progress in image and video processing, network architecture optimization and multi-modal learning. In addition, researchers are committed to designing more accurate adversarial examples to improve the robustness of the model. *Neekhara et al. (2021)* conducted adversarial attacks on deepfake detectors in a black box environment, studied the degree of transfer of adversarial perturbations between different models, and proposed techniques to improve the transferability of adversarial capabilities. The research also generalizes adversarial perturbations to create more accessible attacks, constituting very feasible attack scenarios that can be easily shared among attackers. *Dong et al. (2023)* proposed a practical adversarial attack that does not require any query on the deepfake face images model. The method is built on a surrogate model based on face reconstruction, and then transfers adversarial examples from the surrogate model directly to an inaccessible black-box deepfake model. *Pinhasov et al. (2024)* exploited the power of AI to develop a defensible deepfake detector. This research generates interpretability graphs for a given method, providing visualization of decision-making factors in AI models, employing pre-trained feature extractors to process input images and their corresponding AI images, enhancing understanding of possible adversarial attacks, pinpoint potential vulnerabilities. *Qu et al. (2024)* proposed a robust adversarial perturbation strategy that provides persistent protection for OSN-compressed face images. This study incorporates the trained ComGAN as a sub-module of the target deepfake model, introducing a novel target-level destruction-aware constraint during training. *Khan et al. (2024)* proposed adversarial feature similarity learning, which integrated three basic deep feature learning paradigms to maximize the difference between adversarial perturbed examples and unperturbed examples by optimizing the similarity between samples and weight vectors. The similarity ensures a clear separation between the two categories. *Seow et al. (2023)* believes that existing deepfake

detection methods based on deep learning mainly rely on complex convolutional neural networks. However, these methods have high computational costs and are vulnerable to adversarial attacks. This study proposes a shallower and more cost-effective deep neural networks. _Uddin et al. (2023)_ proposed a robust multi-instance learning method by introducing additional GAN-based operations, exposing GAN-based AF to manipulated images, and then using multiple additional generators to generate multiple real-world AFs from real images. instances, and finally trained collaboratively with multiple real adversarial attack instances in an open-set manner.

### Research on deepfake based on generative adversarial network

In recent years, deepfake research based on generative adversarial networks is focusing on improving the authenticity of generated images or videos, improving generation efficiency, and developing effective detection and defense mechanisms. _Abbas & Taeihagh (2024)_ explores automated key detection and generation methods, frameworks, algorithms, and tools for identifying deepfakes (audio, image, and video), and how these methods can be used in different situations to combat the spread of deepfakes and the generation of false information. Popular research results include improved GAN architectures, such as conditional generative adversarial networks (_Wang et al., 2024_) that introduce conditional variables, recurrent generative adversarial networks (_Li & Wang, 2021_) that allow conversion between different domains, and StyleGAN that can finely control styles and attributes (_Sauer, Schwarz & Geiger, 2022_). These advances not only improve the visual fidelity of generated content, but also increase the model's ability to control specific styles or attributes. _Guarnera, Giudice & Battiato (2024)_ proposed a hierarchical multi-level approach. The first layer classifies real images _versus_ AI-generated images. The second layer distinguishes the images generated by GAN and DM. The third layer implements specific GAN and DM architectures for generating synthetic data. _Kalpokas & Kalpokiene (2022)_ used a generative adversarial network to further shrink the network composed of thousands of units, so that these simple functions can be combined to perform complex deepfake functions such as object recognition. _Aduwala et al. (2021)_ explored GAN discriminator-based solutions as a means of detecting deepfake videos, using MesoNet as a baseline to train GAN and extracting the discriminator as a dedicated module for detecting deepfake. _Li et al. (2023b)_ mainly researched the methods used to implement deepfake, discussed the main deepfake manipulation and detection techniques, and used deep convolution-based GAN models to implement and detect deepfake. _Yang et al. (2021)_ used novel transformation-aware adversarial perturbed faces to defend against GAN-based deepfake attacks. The attack exploits differentiable random image transformations during the generation process. This research also proposes an ensemble-based approach to enhance defense against deepfake variants. To summarize, an overview of existing mainstream research is shown in the Table 1.

## ALGORITHM DESIGN

As far as the current research status is concerned, the training results of many studies can only be targeted at specific models, and the adversarial attack strategies do not have strong
**Table 1 Overview of existing mainstream research.**

| Study | Research focus | Algorithm | Evaluation metrics |
|---|---|---|---|
| Li et al. (2023a), Li et al. (2023b) | Deepfake Generation | Deep convolution-based GAN models | Visual fidelity, authenticity |
| Ali et al. (2024) | Face Manipulation | Deep learning-based face modification | Realism, perceptual quality |
| Kuang et al. (2021) | Face Replacement | Generative Adversarial Network (GAN) | Naturalness, visual quality |
| Wang et al. (2023) | Style Transfer in Face Images | Multi-image style transfer loss function | Realism, style consistency |
| Huang et al. (2023) | Face Swap Detection | Implicit identity-driven framework, CNN | Detection accuracy, identity contrast |
| Han et al. (2023) | Homologous Deepfake Detection | FCD-Net, facial prominence saliency algorithm | Classification accuracy, feature extraction |
| Waseem et al. (2023) | Face and Expression Replacement | Existing deepfake methods | Benchmarking performance |
| Akhtar (2023) | Types of Deepfakes and Detection | Various generation and detection methods | Method coverage, detection accuracy |
| Neekhara et al. (2021) | Adversarial Attacks on Deepfake | Adversarial perturbations transfer | Attack transferability, model robustness |
| Dong et al. (2023) | Practical Adversarial Attack | Surrogate model-based attacks | Attack effectiveness, model vulnerability |
| Pinhasov et al. (2024) | Defensible Deepfake Detection | Interpretability graphs, pre-trained feature extractors | Interpretability, decision factors |
| Qu et al. (2024) | Robust Adversarial Perturbations | ComGAN, target-level destruction-aware constraint | Protection persistence, robustness |
| Khan et al. (2024) | Adversarial Feature Similarity Learning | Adversarial feature similarity learning | Feature separation, classification accuracy |
| Seow et al. (2023) | Efficient Deepfake Detection | Shallower deep neural networks | Computational cost, detection accuracy |
| Uddin et al. (2023) | Robust Multi-Instance Learning | GAN-based operations, multi-instance learning | Robustness, training effectiveness |
| Abbas & Taeihagh (2024) | Key Detection and Generation Methods | Automated key detection, frameworks | Detection effectiveness, applicability across media |
| Wang et al. (2024) | GAN Architecture Improvement | Conditional GAN | Image quality, conditional control |
| Li & Wang (2021) | Domain Conversion with GANs | Recurrent GAN | Domain adaptability, generation quality |
| Sauer, Schwarz & Geiger (2022) | Style and Attribute Control | StyleGAN | Style control, visual fidelity |
| Guarnera, Giudice & Battiato (2024) | Multi-Level Detection Approach | Hierarchical GAN and DM architectures | Detection accuracy, image classification |
| Kalpokas & Kalpokiene (2022) | Network Shrinking with GANs | Network shrinking GAN | Network efficiency, deepfake performance |
| Aduwala et al. (2021) | GAN Discriminator for Detection | GAN discriminator-based detection | Detection accuracy, discriminator performance |
| Yang et al. (2021) | Defense Against GAN-Based Attacks | Transformation-aware adversarial perturbations | Defense effectiveness, perturbation robustness |

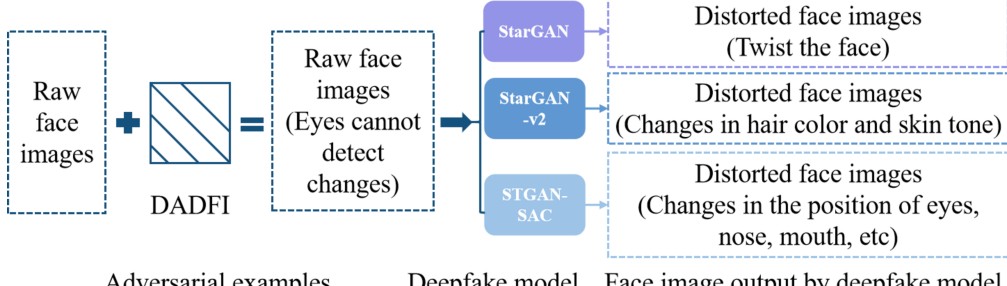

**Figure 1** The effect achieved by DADFI.

effectiveness and transferability. To address this problem, this paper designs a destructive active defense algorithm (DADFI) for deep fake face images. This algorithm can design more effective and more transferable adversarial samples. This type of adversarial samples can make mainstream deep fake face image models (such as StarGAN, StarGAN-v2 (*Cai, Li & Zhang, 2023*) and STGAN-SAC (*Hou & Nayak, 2023*)) output severely distorted fake images. The face image adversarial samples generated using the DADFI are almost indistinguishable from the original images when observed by human vision. However, when these adversarial samples are input into the deepfake face image models, the output face images are greatly distorted. The defense is implemented, as shown in Fig. 1, where the face images come from the public datasets CASIA-FaceV5 (*Qi et al., 2023*) containing 2,500 face images from 500 Asians and CelebA.

The destructive active defense algorithm (DADFI) designed in this paper first generates adversarial samples, and then proposes an optimization algorithm for fine-tuning the adversarial samples to improve the attack, effectiveness, and transferability.

## Strategies for adversarial samples generation and fine-tuning

This algorithm innovatively designs an adversarial samples generation module and an adversarial samples fine-tuning module. In the generation module, a generator $Gen$ is designed, and in the fine-tuning module, two recognizers $Ide_x$ and $Ide_y$ are designed. In addition, $n$ deepfake models $mod_i$ are set, which $i \leq n$. If any face image is set as $image_i$, the feature space of the face image is set as $fea_{image_i}$, and the number of all features is set as $sum_{fea}$. Next, the generator $Gen$ extracts the high-dimensional feature $Gen_{image_i}$ of the face image $image_i$, and superimposes the feature value to the original face image to obtain the adversarial sample $iamge_{i_{adv}}$ of the face image, which is composed of a set of feature vectors, that is, $iamge_{i_{adv}} = image_i + Gen_{image_i}$. Based on the projected gradient descent algorithm, a basic adversarial perturbation $adv_i$ is generated for the deepfake model $mod_i$. This adversarial perturbation is added to the original face image to obtain the fine-tuned adversarial sample $image'_{i_{adv}}$ of the face image. The adversarial sample is input into the image output by the deepfake model. Produces obvious distortion, that is, $image'_{i_{adv}} = image_i + adv_i$.

In order to obtain adversarial samples that are transferable and universal across models, this paper innovatively uses the projected gradient descent algorithm to implement adversarial attacks based on gradients. The distortion area and distortion effect caused by the basic adversarial perturbation $adv_i$ on the face image is only the same as the corresponding deepfake models are related, that is to say, this basic adversarial perturbation is robust to multiple deepfake models. In summary, the high-dimensional feature $Gen_{image_i}$ has learned adversarial attack capabilities similar to the basic adversarial perturbation $adv_i$, and the adversarial samples have characteristics that are common across models.

This algorithm sets $mod_i(x)$ as the face editing operation of the selected deepfake model, sets the deepfake image generated by the adversarial sample $image_{i_{adv}}$ of the face image as $fake_{image_i} = mod_i(image_{i_{adv}})$, and sets the deepfake image generated by the adversarial sample $image'_{i_{adv}}$ based on fine-tuning of the face image as $fake'_{image_i} = mod_i(image'_{i_{adv}})$, input $y_i^{fake} = fake_{image_i}$ and $y_i^{real} = fake'_{image_i}$ into the recognizer $Ide_y$, respectively, and improve the attack capability of the adversarial sample $image_{i_{adv}}$ through adversarial training. In addition, two types of adversarial samples are input into $n$ different deepfake models, and based on the output results, a distorted deepfake face images set $[(fake_{image_1}, fake'_{image_1}), (fake_{image_2}, fake'_{image_2}), \ldots, (fake_{image_n}, fake'_{image_n})]$ is constructed. Joint training through discriminators can improve the transferability of adversarial examples. Based on the above strategy, inputting the original face images into the trained generative model can obtain adversarial samples. Compared with the original images, the adversarial samples are basically indistinguishable to the human eye. However, these samples can completely make several different deepfake models output face images with different degrees of distortion, as shown in Fig. 2.

## Algorithm for generating adversarial examples

When generating adversarial examples, traditional adversarial example attack algorithms need to repeatedly access the deepfake model. In order to solve this problem and improve the efficiency of adversarial sample generation, this paper innovatively uses a generative adversarial network to design an adversarial sample generation module and an adversarial sample identification module. The generation module designs the generator, and the identification module designs the recognizer. In this network architecture, the input is the original face image, and the output is the generated adversarial sample.

The generator Gen is used to generate a slight perturbation that can make the adversarial sample image similar to the original face image, and then obtain the mapping function from the original face image to the slight perturbation as shown in Eq. (1)

$$Gen : image_i \longrightarrow Gen_{image_i}. \tag{1}$$

Among them, $Gen$ represents the generator, $image_i$ represents any face image input to the generator, and $Gem_{image_i}$ represents the slight adversarial perturbation output by the generator. The adversarial sample $image_{i_{adv}}$ can be obtained by superimposing the perturbation onto the original image, as shown in Eq. (2):

$$image_{i_{adv}} = image_i + Gen_{image_i}. \tag{2}$$

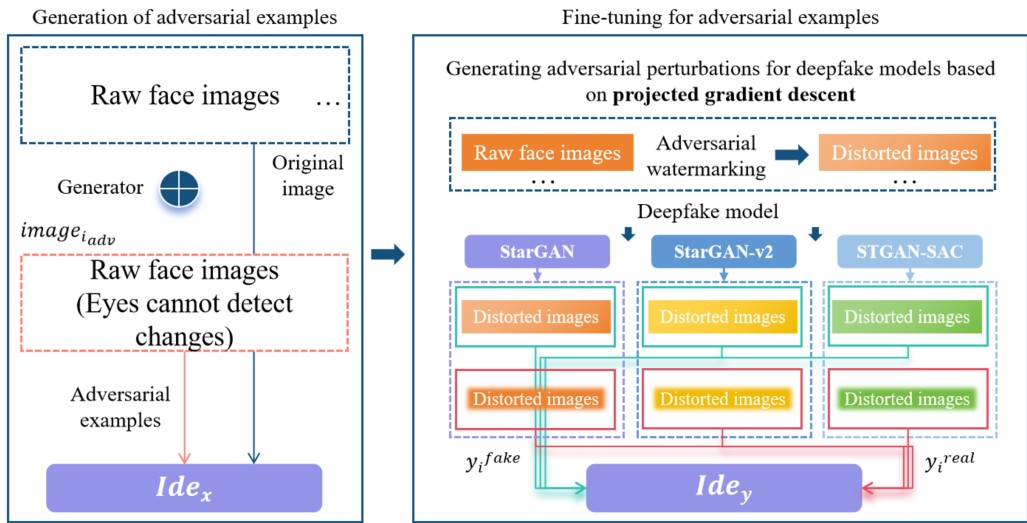

**Figure 2  Adversarial example generation and fine-tuning.**

After obtaining the adversarial samples, this algorithm uses the recognizer $Ide_x$ to distinguish them in order to reduce the visual gap between the adversarial samples and the original face images. This algorithm innovatively proposes adversarial training to reduce the visual impact caused by slight perturbations, making the adversarial samples almost visually consistent with the original face images. The generator defined by the algorithm adopts an autoencoding structure. The encoder uses three layers of convolution, and instance normalization is performed after each layer of convolution. The decoder uses three upsampling layers, and instance normalization is performed after each layer of convolution. In addition, four residual blocks are added between the encoder and the decoder. Each residual block consists of two 3*3 convolutional layers. At the same time, instance normalization is performed after each convolutional layer. The first of the residual blocks is the output of the convolutional layer is activated.

The recognizer $Ide_x$ is used to distinguish whether the generated adversarial samples are real or fake compared to the original face images, and iterate during adversarial training so that the generator finally generates a perturbation with minimal visual impact on the original images. The algorithm sets the adversarial loss of the generator *Gen* and the discriminator $Ide_x$ to be $Loss_{Ide_x}$, as shown in Eq. (3):

$$Loss_{Ide_x} = E[logIde_x(image_i)] + E[log(1 - Ide_x(image_i + Gen_{image_i}))] \tag{3}$$

where E represents the mathematical expectation value. At the same time, in order to improve the perception of adversarial samples and minimize the visual impact of slight adversarial perturbations on the original face images, the algorithm innovatively designs a hinge loss function $Loss_g emel$ to constrain the scale of the perturbation, as shown in Eq. (4)

$$Loss_{gemel} = E_{image_i} max(0, \| Gen_{image_i} \|_{\infty} - lim). \tag{4}$$

Among them, *lim* represents the specified infinite norm limit against perturbation. If $\|Gen_{image_i}\|_\infty$ exceeds *lim*, this function will constrain the perturbation.

## Algorithm for fine-tuning against adversarial examples

The adversarial samples fine-tuning algorithm consists of the deepfake model $mod_i$ and the recognizer $Ide_y$, where the network structure of the recognizer $Ide_y$ is consistent with that of the recognizer $Ide_x$. The input of this algorithm consists of two types of adversarial samples, one is the adversarial samples $image_{i_adv}$ generated by the generator, and the other is the adversarial sample $image_{i_adv}{'}$ added with the basic adversarial perturbation. Among them, the addition of basic adversarial perturbation $adv_i$ can significantly distort the face image output by the deepfake model, as shown in Eqs. (5) and (6):

$$adv_{image_0} = image_{ori} + adv_i. \tag{5}$$

Among them, $adv_{image_0}$ represents the first slight perturbation to the original face image, $image_{ori}$ represents the original face image, and $adv_i$ represents slight perturbation.

$$adv_{image_{i+1}} = sli_{image_{ori}}\{adv_{image_i} + \partial(\nabla_{image_{ori}} L(mod_i(adv_{image_i}), mod_i(image_{ori})))\}. \tag{6}$$

Among them, $adv_{image_{i+1}}$ represents the next iteration of the i-th slight perturbation $adv_{image_i}$ of the original face image. The slight perturbation obtained by iterative training of the above algorithm is universal for the deepfake model. That is to say, not only the few face images, but also all the face images in the training data set can be made to have slight perturbations. The fake model outputs distorted face images. This algorithm implements adversarial attacks based on gradients through the projected gradient descent algorithm in advance, and generates slight perturbations for multiple deepfake models. The perturbed adversarial samples can be used as label-assisted optimization of adversarial samples $image_{i_adv}$.

Adversarial examples fine-tuning uses an adversarial loss function to improve the aggressiveness of adversarial examples. The generation process of slight perturbation is related to the model gradient. That is to say, after different face images are subjected to the same slight perturbation, the deepfake models are input, and the output deepfake images are similar in terms of distortion range and degree of distortion. For this reason, input the fine-tuned adversarial sample $image_{i_adv}{'}$ into the image obtained by the deepfake model $mod_i$, that is, the fake image $fake_{image_i}{'}$ generated based on the fine-tuned adversarial sample $image_{i_adv}{'}$ of the face image is generated; and input the adversarial sample $image_{i_adv}$ generated by the generator. The image obtained by the deepfake model $mod_i$ is obtained, that is, the deepfake image $fake_{image_i}$ generated by the adversarial sample $image_{i_adv}$ of the face image is obtained. At this time, the algorithm identifies the authenticity of the image through the recognizer $Ide_y$, and adds adversarial training to make $fake_{image_i}$ infinitely close to $fake_{image_i}{'}$, and finally the adversarial resistance of the adversarial sample $image_{i_adv}$ is improved. The adversarial loss $Loss_{Ide_y}$ of the recognizer $Ide_y$ is shown in Eq. (7)

$$Loss_{Ide_y} = E[logIde_y(mod_i(image_{i_{adv}}{'}))] + E[log(1 - Ide_y(mod_i(image_{i_{adv}})))]. \tag{7}$$

Adversarial examples fine-tuning integrates multiple deepfake models to improve the transferability of adversarial examples. During the training process, this algorithm

uses adversarial samples to attack multiple deepfake models, which not only improves transferability, but also improves the cross-model attack capability of adversarial samples. First, this algorithm generates basic adversarial perturbations $adv_i(i \in [1, n])$ for $n$ deepfake models. Then, the adversarial samples $image_{i_{adv}}$ and adversarial samples $image'_{i_{adv}}$ are input into multiple different deepfake models respectively, where the adversarial samples $image_{i_{adv}}$ obtains the deepfake images $fake_{image_i}$ through the deepfake model, and the adversarial samples $image'_{i_{adv}}$ obtain the deepfake images $fake'_{image_i}$ through the deepfake model, and then collect these images as $ass$, as shown in Eq. (8):

$$ass = [(fake'_{image_1}, fake_{image_1}), (fake'_{image_2}, fake_{image_2}), \ldots, (fake'_{image_n}, fake_{image_n})]. \tag{8}$$

This algorithm makes $fake_{image_i}$ visually closer to $fake'_{image_i}$ through adversarial training, and can further improve the loss function, as shown in Eq. (9):

$$Loss_{Ide_y} = E[\sum_{i=1}^{n} logIde_y(mod_i(image'_{i_{adv}}))] + E[\sum_{i=1}^{n} log(1 - Ide_y(mod_i(image_{i_{adv}})))]. \tag{9}$$

After the above improvement process, the objective function $Loss$ of this algorithm is composed of the adversarial loss $Loss_{Ide_x}$ of the recognizer $Ide_x$, the adversarial loss $Loss_{Ide_y}$ of the recognizer $Ide_y$ and the hinge loss $Loss_{gemel}$, as shown in Eq. (10):

$$Loss = Loss_{Ide_x} + Loss_{Ide_y} + Loss_{gemel} \tag{10}$$

where μ represents the weight of the hinge loss. During the training of this algorithm, the adversarial loss $Loss_{Ide_x}$ constrains the adversarial samples and the original face images to be highly consistent visually, and the hinge loss $Loss_{gemel}$ constrains the boundaries of slight disturbances. The purpose of both losses is to ensure the generation of high quality. , adversarial samples with visual effects close to the original face images. Deepfake is generally based on a black box scenario. The party whose image is deeply faked does not know the network structure and parameters of the deepfake model. Therefore, the best way to proactively defend is to perform black box attacks on several deepfake models at the same time. This algorithm attacks based on queries, and estimates the gradient of the model by querying the difference in the output results of the deepfake model to improve the model's generalization ability. Adversarial loss $Loss_{Ide_y}$ simulates adversarial samples to attack multiple deepfake models in a black box scenario, thereby improving the aggressiveness and transferability of adversarial samples.

## Strategies to improve the offensiveness of adversarial examples

In traditional research on increasing the aggressiveness of adversarial samples, most studies use fine-tuning strategies. The algorithm proposed in this paper innovatively uses two strategies to increase or maintain the aggressiveness of samples. These two strategies are using a stronger perturbation strategy and dynamically adjusting the perturbation intensity.

### Stronger perturbation strategy

When designing a stronger perturbation strategy, this paper adopts a method of gradually accumulating small perturbations. First, the basic adversarial perturbation $adv_i$ is generated

and superimposed on the original face image, as shown in Eq. (11):

$$adv_{image_0} = image_{ori} + adv_i. \tag{11}$$

Among them, $adv_{image_0}$ represents the first slight perturbation of the original face image, $image_{ori}$ represents the original face image, and $adv_i$ represents a slight perturbation. Then, a stronger perturbation is iteratively trained, as shown in Eq. (12):

$$adv_{image_{i+1}} = sli_{image_{ori}} adv_{image_i} + \partial(\nabla_{image_{ori}} L(mod_i(adv_{image_i}), mod_i(image_{ori}))). \tag{12}$$

Among them, $adv_{image_i}$ represents the next iterative perturbation of the original face image. Through multiple iterations, we are able to generate universal small perturbations applicable to different deep fake models, so that the face images in all training datasets have small perturbations, which causes the deep fake model to output distorted face images. This perturbation strategy generates small perturbations through the projected gradient descent algorithm to conduct adversarial attacks on multiple deep fake models, improving the universality and attack effect of the perturbation.

### Adjust perturbation intensity

In order to improve the offensiveness, this paper designs a method to dynamically adjust the perturbation strength. In the process of adversarial sample fine-tuning, this paper introduces the adversarial loss function $Loss_{Ide_y}$, which estimates the gradient of the model by querying the output difference of the deep fake model, so as to conduct adversarial attacks in black box scenarios, as shown in Eq. (13):

$$Loss_{Ide_y} = E[logIde_y(mod_i(image'_{iadv}))] + E[log(1 - Ide_y(mod_i(image_{iadv})))]. \tag{13}$$

During the training process, the refined adversarial sample $image'_{i_{adv}}$ and the generated adversarial sample $image_{i_{adv}}$ are input, and distorted images are obtained through multiple different deep fake models. Through adversarial training, $fake_{image_i}$ is made close to $fake'_{image_i}$, and the loss function is further optimized, as shown in Eq. (14):

$$Loss = Loss_{Ide_x} + Loss_{Ide_y} + Loss_{gemel} \tag{14}$$

where $Loss_{gemel}$ represents the weight of the hinge loss. Through the constraints of the adversarial loss $Loss_{Ide_x}$ and the hinge loss $Loss_{Ide_y}$, high-quality adversarial samples are generated to make them visually close to the original face images. In the black-box attack scenario, this method of dynamically adjusting the perturbation strength can improve the aggressiveness and transferability of adversarial samples, thereby enhancing the ability to attack multiple deep fake models.

Based on the above algorithm design, the ultimate goal of this algorithm is to generate adversarial samples that are almost visually consistent with the original face images. At the same time, after the adversarial samples are input to different deepfake models, highly distorted face images are output. This algorithm also innovatively uses a generative adversarial network for optimization. The purpose of the confrontation of the generator *Gen* is to make the target loss function as small as possible, and the purpose of the confrontation of the recognizer $Ide_x$ and $Ide_y$ are to make the target loss function as large as possible, and finally reach Nash equilibrium to end the adversarial training.

## EXPERIMENTS

### Models, datasets and parameters

The deepfake models selected for the experiments of this paper are $StarGAN$, $StarGAN - v2$ and $STGAN - SAC$. All three models can achieve deepfake of face images. Among them, $StarGAN$ and $StarGAN - v2$ are used for deepfake of hair and age in face images, while $STGAN - SAC$ is used for deepfake of accessories in face images.

In addition, the datasets selected for the experiment are the public datasets $CASIA - FaceV5$ and $CelebA$. Among them, the $CASIA - FaceV5$ dataset contains a large number of high-definition, diverse and real face images, covering different ages, genders and expressions, providing a comprehensive foundation for face recognition and analysis tasks. The $CelebA$ dataset is another large-scale face attribute dataset, containing more than 200,000 celebrity images, each of which is annotated with 40 attribute labels. The images in this dataset have a wide range of variations in pose, background and lighting conditions. The algorithm proposed in this paper novelly adds the generative adversarial network to the experiment. In the experiment, images of size $300 * 300$ extracted from the dataset are used as input face images. The weight $\mu$ representing the hinge loss is set to 0.5. The training cycle and batch processing are set to 100 and 50, respectively.

Finally, the algorithm innovatively designed a model optimizer for the experiments. The learning rate of each parameter is dynamically adjusted by calculating the first and second moment estimates of the gradient, thereby improving training efficiency and accuracy. The specific steps are:

1. Compute gradient $gra_{para}$ for each parameter $para$.
2. Calculate the first-order moment estimate mean $\overline{est_1}$ of the gradient, as shown in Eq. (15):
$$\overline{est_1} = \alpha_1 \overline{est_1} + (1 - \alpha_1) gra_{para}. \tag{15}$$
   Among them, $\alpha_1$ is used to control the exponential decay rate of the first-order moment estimate, which is set to 0.9 in the experiments.
3. Calculate the second-order moment estimation variance $\sigma$ of the gradient, as shown in Eq. (16):
$$\sigma = \alpha_2 \sigma + (1 - \sigma) gra_{para}^2. \tag{16}$$
   Among them, $alpha_2$ is used to control the exponential decay rate of the second-order moment estimation, which is set to 0.99 in the experiments.
4. Perform bias correction on the first-order moment estimate and the second-order moment estimate, as shown in Eq. (17):
$$cor_{\overline{est_1}} = \frac{est_1}{1 - \alpha_1^i}, cor_\sigma = \frac{\sigma}{1 - \alpha_2^i}. \tag{17}$$
   Among them, $i$ represents the current iteration number.
5. Update parameters based on first-order moment estimation and second-order moment estimation, as shown in Eq. (18):
$$para = para - \gamma \frac{cor_{\overline{est_1}}}{\sqrt{cor_\sigma} + \tau}. \tag{18}$$
   Among them, $\gamma$ represents the learning rate, and $\tau$ is used to stabilize the value. In the experiments, it is set to $1e - 6$.

Based on the above, the model optimizer designed in the experiments have a good convergence speed.

## Measurement methods for experimental results

First, the experiments innovatively designed the pixel distance $d_{pix}$. When evaluating the quality of the face images generated by the deepfake model, the quality of the generated images can be measured by calculating $d_{pix}$ between the original face images and the deepfake face images, as shown in Eq. (19):

$$d_{pix} = \sqrt{\sum_{i=1}^{n}(I_{input}[i] - I_{output}[i])^2}. \tag{19}$$

Among them, $I_{input}[i]$ represents the value of the $i-th$ pixel of the input original face images, and $I_{output}[i]$ represents the value of the $i-th$ pixel of the output deepfake face images. It is not difficult to realize that the larger the value of $d_{pix}$, the greater the visual difference between the output deepfake face images and the original face images, that is, the better the attack effect of the adversarial samples.

Then, the value of $d_{pix}$ can more accurately evaluate the quality of the face images generated by the deepfake model from a global level. However, if the visual difference between the output deepfake images and the original face images is mainly local, it should also be evaluated. It is evaluated that the attack effect is good, but the $d_{pix}$ value at this time may be very small. At this time, the experiments innovatively adds a mask matrix to calculate the mask distance. The experiments use an attention priority mechanism to focus attention on the modified parts of the face images, because the mask matrix can specify which part of the pixels should be used to calculate distance, and which part of the pixels should be ignored. If the difference in this part of the area exceeds a certain threshold, it is also considered to be a good attack effect. Calculate the mask distance $d_{M_{d\_pix}}$ as shown in Eq. (20):

$$d_{M_{d\_pix}} = \sqrt{\sum_{i=1}^{n}M[i](I_{input}[i] - I_{output}[i])^2}. \tag{20}$$

Among them, $M$ is the mask matrix defined in this experiment. The shape of the matrix and the original face image are both $300 * 300$, in which the value of each pixel is 0 or 1.0 means that the pixel at the position should be ignored, and 1 means that the pixel at the position should be used to calculate the distance. $M[i]$ is the value of the $i-th$ pixel in the mask matrix.

Finally, in order to further evaluate the effect of attacking the deepfake model, the experiments will be conducted by calculating the *Frechet* distance $d_{Frechet}$ using the mean vector and covariance matrix of the real images and the generated images (*Cheng & Huang, 2023*), as shown in Eq. (21):

$$d_{Frechet} = \left\| \mu_{input} - \mu_{output} \right\|_2^2 + Tr(\sum input + \sum output - 2\sqrt{\sum input \sum output}). \tag{21}$$

The lower the distance $d_{Frechet}$ value, the closer the distribution of the generated images and the real images in the feature space are, and the higher the quality of the generated images are. Among them, $\mu_{input}$ and $\mu_{output}$ represent the mean vector of the real images and the generated images in the feature space respectively, $\sum input$ and $\sum output$ represent the covariance matrix of the real images and the generated images in the feature space respectively, and $\|\mu_{input} - \mu_{output}\|_2^2$ calculates the covariance matrix of the real images and the generated images in the feature space. The square of the Euclidean distance of the mean vector is used to measure the difference in mean between the two. $Tr(\sum input + \sum output - 2\sqrt{\sum input \sum output})$ calculates the *Frechet* distance between the covariance matrix of the real images and the generated images in the feature space to measure the difference in covariance between the two. Finally, the distances between these two parts are added to obtain the *Frechet* distance value, which is used to evaluate the quality of the generated images.

## Results

### Validity verification

Based on the results of the experiments, this paper designed an objective method to evaluate whether the attack deepfake model was successful. The experiments randomly selected 200 face images from the public datasets $CASIA - FaceV5$ and $CelebA$ as the training set, and randomly selected 2000 face images from other images in the dataset for all experiments. By calculating the pixel distance $d_{pix}$, mask distance $d_{M_{d\_pix}}$ and *Frechet* distance $d_{Frechet}$, we observe the active defense effect against the deepfake models $StarGAN$, $StarGAN - v2$ and $STGAN - SAC$, as shown in Fig. 3.

First, this experiment randomly selected 200 face images from the dataset $CASIA - FaceV5$ for testing, and modified these original face images based on the face deepfake characteristics of the deepfake models $StarGAN$, $StarGAN - v2$ and $STGAN - SAC$. The results show that the pixel distance $d_{pix}$ of the three models attacked by the DADFI algorithm proposed in this paper is greater than 0.06. The algorithm believes that when the $d_{pix}$ of the original face images and the deepfake images are greater than 0.6, people can easily distinguish the two images visually. In other words, when the $d_{pix}$ of the two images is greater than 0.6, the active defense effect is good. In addition, the mask distance $d_{M_{d\_pix}}$ of the three models attacked by the DADFI algorithm is greater than 0.85. The closer the mask distance $d_{M_{d\_pix}}$ is to 1, the better the active defense effect. At the same time, the distorted *Frechet* distance $d_{Frechet}$ values of all output images are above 30, indicating that the difference between distorted images and normal deepfake images are large, and also prove the effectiveness of active defense. The same experiment was also conducted on the $CelebA$ dataset and achieved similar results.

Then, during the training process of the data set, the DADFI algorithm constrains the infinite norm limit *lim* representing the specified adversarial perturbation to less than 0.05, so there is basically no difference between the original face images and its adversarial samples for normal vision. At the same time, there are large visual differences between the output results of the deepfake models $StarGAN$, $StarGAN - v2$ and $STGAN - SAC$ on the original face images and its adversarial samples, human vision can easily detect tampered

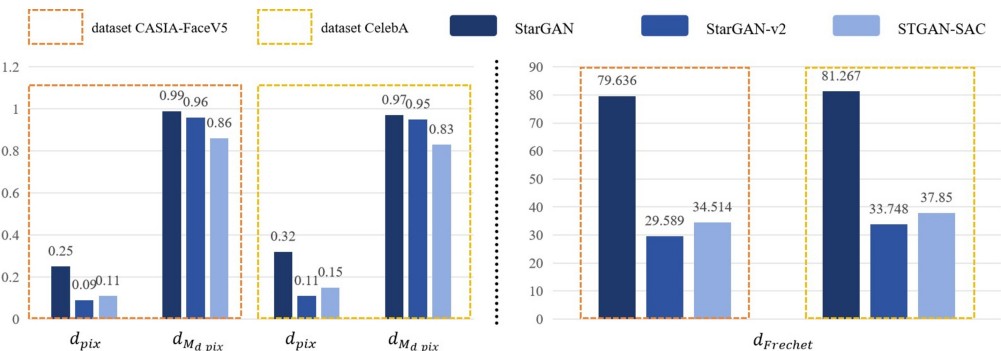

**Figure 3 Experimental results.**

images. Experimental results show that *StarGAN* is most vulnerable to adversarial attacks, and the output results after being attacked have the most obvious distortion. Although *StarGAN − v2* and *STGAN − SAC* have a certain degree of robustness to attacks, the output images after the attack are still significantly different from the normal deepfake images, and the models cannot achieve the purpose of tampering with face images.

Finally, by horizontally comparing and modifying different facial feature attributes of the same models, it was found that the tampered area of the deepfake model will not have a major impact on the generated distortion area, that is, the adversarial attack is robust to deepfake methods. The above experimental results show that the adversarial samples generated by DADFI have the ability to attack deepfake models across models and can protect face images from tampering by deepfake models.

### Comparative experiments

First, based on the datasets *CASIA − FaceV5* and *CelebA*, the algorithm selected three common adversarial attack algorithms for comparison. They are *MOMA* proposed by *Sun et al. (2023a)*, *ATS−O2A* proposed by *Li et al. (2023a)* and *Adv − Bot* proposed by *Debicha et al. (2023)*. Based on the above three algorithms, the experiments innovatively use mask distance $d_{M_{d\_pix}}$ and *Frechet* distance $d_{Frechet}$ to observe the active defense effects of several different algorithms against deepfake models *StarGAN*, *StarGAN − v2* and *STGAN − SAC*. The experimental results are as follows as shown in Figs. 4, 5 and 6.

Then, the experimental results conducted on three different deepfake models show that the above three common adversarial attack algorithms have certain active defense effects. Compared with the comparative algorithms, the DADFI proposed in this paper can generate adversarial samples that can more effectively attack the two deepfake models of *StarGAN* and *STGAN–SAC*, achieving the goal of cross-model defense against deepfake. In addition, the maximum value of the mask distance $d_{M_{d\_pix}}$ of the DADFI for attacking these three models is 1.00 for *StarGAN*, and the minimum value is 0.79 for *STGAN–SAC*, but it is far ahead of the values of other algorithms. Because the closer the mask distance $d_{M_{d\_pix}}$ is to 1, the better the active defense effect is, so the active defense effect of DADFI is better than the comparison algorithms on these three models. At the same time, the

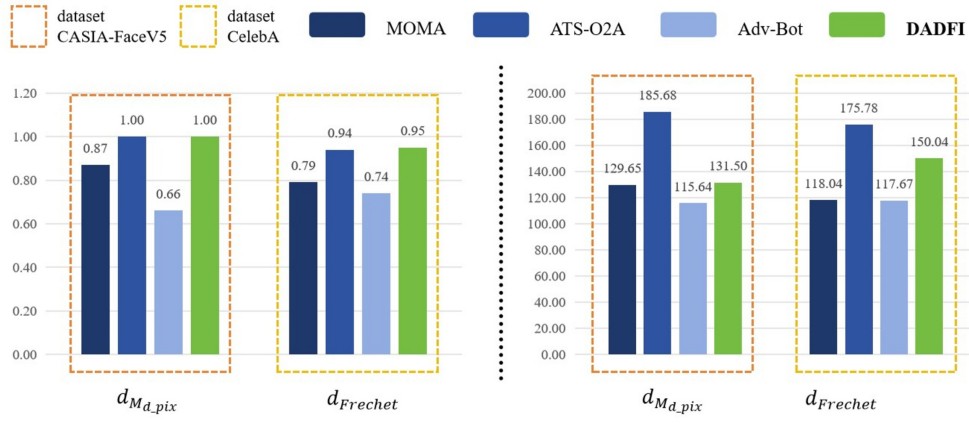

**Figure 4** StarGAN.

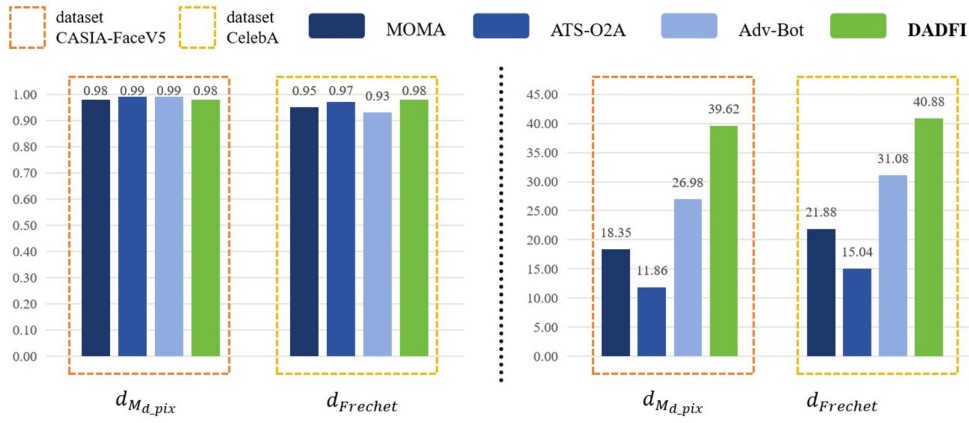

**Figure 5** StarGAN-v2.

distorted *Frechet* distance $d_{Frechet}$ value of all output images of the DADFI is above 30, with the maximum value being 131.50 for *StarGAN* and the minimum value being 39.62 for *StarGAN − v2*, indicating that the difference between the distorted images and the normal deepfake images are large, and also prove effectiveness of active defense.

Finally, the DADFI proposed in this paper is based on an end-to-end generative adversarial network. After training, adversarial samples can be obtained without accessing the deepfake model again. Therefore, this paper compares the efficiency of adversarial samples generation with different algorithms. The experimental results are as shown as Fig. 7.

Experimental results show that compared with the comparative algorithms *MOMA, ATS − O2A* and *Adv–Bot*, the experimental results of the DADFI are better in terms of adversarial samples generation efficiency.

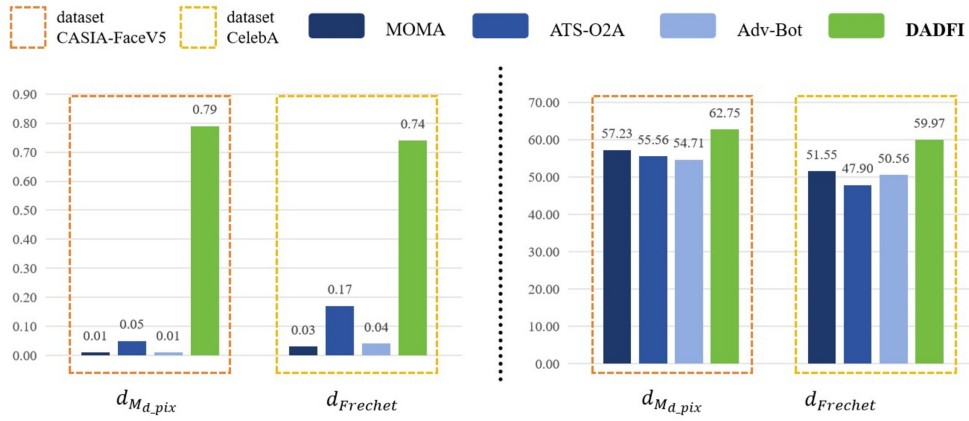

**Figure 6   STGAN-SAC.**

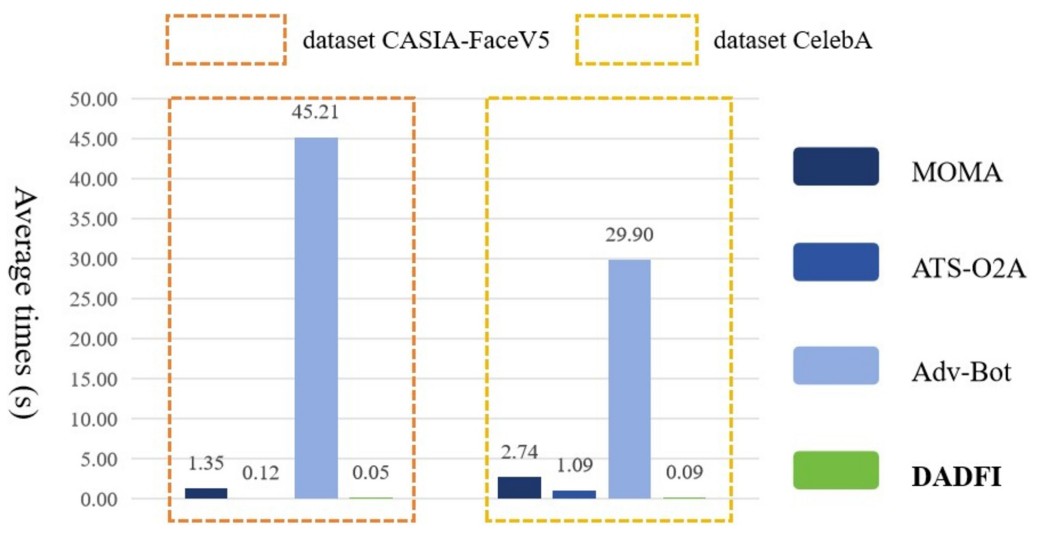

**Figure 7   The average generation time of adversarial examples.**

## Verification experiments

In order to verify the aggressiveness and migration of the DADFI proposed in this paper, the paper designed the following two sets of experiments.

**Verification experiment 1:** Adjust the number of deepfake models in the adversarial samples fine-tuning algorithm, and adjust the three deepfake models trained for adversarial samples attack in DADFI to attack a single deepfake model, in order to verify the effectiveness of the DADFI and the advantages in mobility.

Adjust the number of deepfake models to 1, and when only one of the deepfake models *StarGAN*, *StarGAN−v2*, and *STGAN−SAC* is retained, the attack and transferability of the adversarial samples are evaluated through pixel distance $d_{pix}$. If the pixel distance $d_{pix}$ produced when attacking the selected deepfake model is high, and the pixel distance

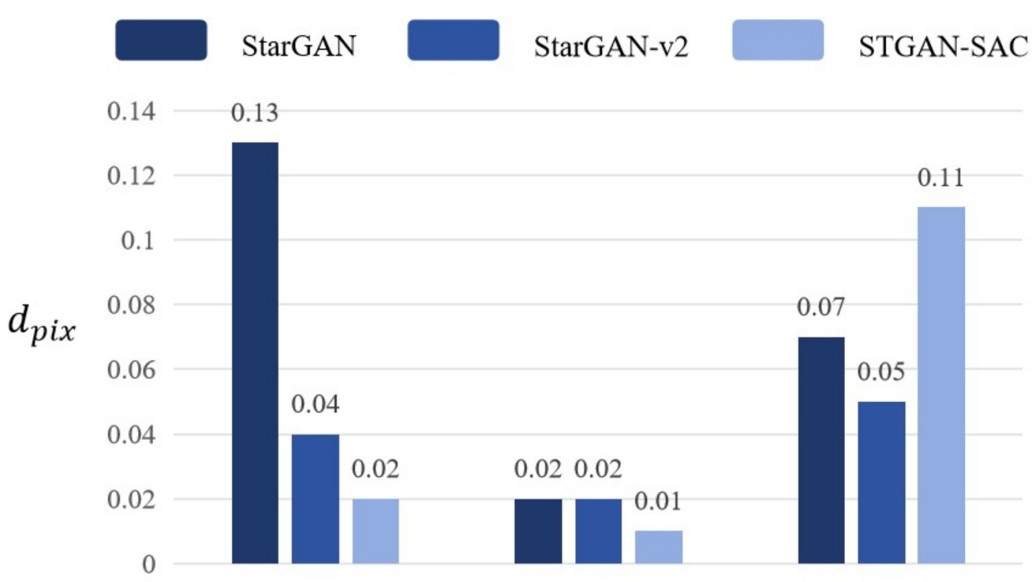

**Figure 8** Pixel distance comparison.

$d_{pix}$ produced when attacking the remaining deepfake models is significantly lower than 0.05, in other words, in the case of a single-model attack, the pertinence is high, but the transferability is insufficient. Experimental results prove that the DADFI proposed in this paper uses three deepfake models for training, and uses pixel distance $d_{pix}$ loss to evaluate the distortion effect. The values are all higher than 0.05, and the mask distance $d_{M_{d\_pix}}$ scores are high, both close to 1, which is obviously improved transferability of adversarial examples. The pixel distance comparison of any one of the three deepfake models is retained as shown in Fig. 8.

Experimental results show that the DADFI proposed in this paper adjusts the number of deepfake models to 1. When only one of the deepfake models $StarGAN$, $StarGAN-v2$ and $STGAN-SAC$ is retained, the pixel distance values corresponding to this model are higher than compare models to prove that the transferability of adversarial examples is better.

**Verification experiment 2:** Remove the adversarial samples fine-tuning algorithm, and determine whether the adversarial samples generated by the generator can make the deepfake model output distorted face images to verify whether the fine-tuning algorithm can improve the aggressiveness of the adversarial samples.

When the adversarial samples fine-tuning algorithm is removed from training and only the adversarial samples generation algorithm is used for training, the pixel distance $d_{pix}$ of the generated adversarial samples to the deepfake model $StarGAN$ is only 0.0382, and the pixel distance $d_{pix}$ of the other two deepfake models is even lower. Moreover, the mask distance $d_{M_{d\_pix}}$ values of the above three models are all small, and the adversarial samples are almost not aggressive. By comparing with the DADFI proposed in this paper,

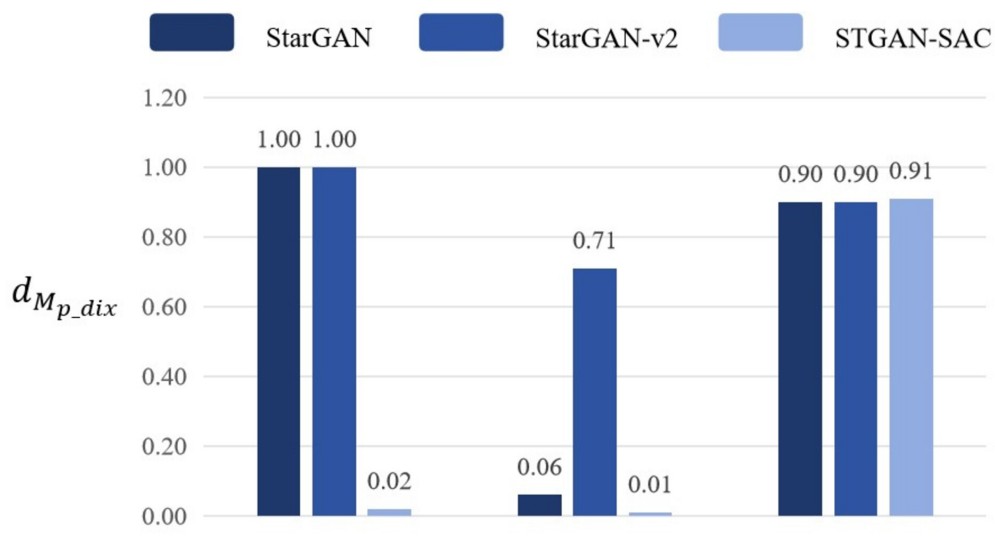

**Figure 9** Mask distance comparison.

DADFI can effectively improve the aggressiveness of adversarial samples and retain the mask distance comparison of any one of the three deepfake models as shown in Fig. 9.

Experimental results show that when the adversarial samples fine-tuning algorithm is removed from the DADFI training proposed in this paper and only the adversarial samples generation algorithm is retained, the mask distance $d_{M_{d\_pix}}$ values corresponding to the model are higher than those of the comparison models, thus proving the attack performance comparison of adversarial samples good.

The results of the above-mentioned verification experiment one and verification experiment two show that the DADFI proposed in this paper has strong attack and migration performances.

## CONCLUSION

This paper proposes a destructive active defense algorithm for deepfake face images (DADFI). Active defense protects face images from being tampered with by deepfake models. The algorithm consists of two sub-algorithms: the adversarial samples generation algorithm and the adversarial samples fine-tuning algorithm. When the adversarial samples generation algorithm generates adversarial samples through generator training, the loss function is used to make the perturbed adversarial samples maintain visual consistency with the original face images, reducing the impact of perturbation on image quality. The adversarial samples fine-tuning algorithm combines existing adversarial attack algorithms to improve the aggressiveness and transferability of adversarial samples through adversarial training, allowing multiple deepfake models to output highly distorted face images. Experimental results show that the DADFI algorithm proposed in this paper can achieve the ability to interfere with the output of deepfake models across models, and the achieved face images distortion effect is close to the adversarial samples generated by the current

mainstream adversarial attack algorithms. More importantly, DADFI greatly improves the efficiency of adversarial samples generation. However, despite the promising results, the research does have some limitations in terms of datasets and models. The datasets used for generating and fine-tuning adversarial samples might not cover the full spectrum of real-world face variations, potentially limiting the generalizability of the algorithm. Additionally, the models used in the experiments may not encompass all types of deepfake generation techniques, which means the DADFI algorithm's performance could vary when faced with unseen or more advanced deepfake models.

Based on the algorithm proposed in this paper, in future work, the authors will solve the following three problems: (1) Study the generation of adversarial samples by local perturbation to reduce the impact on the quality of the original image. (2) Improve the visual consistency of adversarial samples and enhance their similarity with the original image. (3) Optimize the efficiency of adversarial sample generation and improve the practicality of the algorithm.

### Funding
This research was funded by the National Social Science Fund of China (grant no. 22BSH025), the National Natural Science Foundation of China (grant no. 62206241) and the Key Research and Development Program of Zhejiang Province, China (grant no. 2021C03138), the Medium and Long-Term Science and Technology Plan for Radio, Television, and Online Audiovisuals (grant no. 2022AD0400). The funders had no role in study design, data collection and analysis, decision to publish, or preparation of the manuscript.

### Grant Disclosures
The following grant information was disclosed by the authors:
National Social Science Fund of China: 22BSH025.
National Natural Science Foundation of China: 62206241.
Key Research and Development Program of Zhejiang Province: 2021C03138.
Medium and Long-Term Science and Technology Plan for Radio, Television, and Online Audiovisuals: 2022AD0400.

### Competing Interests
The authors declare there are no competing interests.

### Author Contributions
- Yang Yang conceived and designed the experiments, performed the experiments, analyzed the data, performed the computation work, prepared figures and/or tables, authored or reviewed drafts of the article, and approved the final draft.
- Norisma Binti Idris analyzed the data, prepared figures and/or tables, and approved the final draft.

- Chang Liu conceived and designed the experiments, performed the experiments, performed the computation work, prepared figures and/or tables, authored or reviewed drafts of the article, and approved the final draft.
- Hui Wu performed the experiments, analyzed the data, authored or reviewed drafts of the article, and approved the final draft.
- Dingguo Yu conceived and designed the experiments, analyzed the data, performed the computation work, authored or reviewed drafts of the article, and approved the final draft.

### Data Availability

The CASIA-FaceV5 dataset is available from the Institute of Automation Chinese Academy of Sciences at: http://english.ia.cas.cn/db/201610/t20161026_169405.html, and at FigShare: Yang, Yang (2024). CASIA-FaceV5.zip. figshare. Dataset. https://doi.org/10.6084/m9.figshare.26509591.v1.

The CelebA large-scale face dataset is available at: https://mmlab.ie.cuhk.edu.hk/projects/CelebA.html

The code (original code (DADFI) & 3rd party models (StarGAN, StarGAN-v2 and STGAN-SAC)) are available at Zenodo:Yang. (2024). A-destructive-active-defense-algorithm-for-deepfake-face-images [Data set]. Zenodo. https://doi.org/10.5281/zenodo.13283088.

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
