# Peer review of "A destructive active defense algorithm for deepfake face images"

_PeerJ Computer Science, doi:10.7717/peerj-cs.2356_

## Round 0.1 · original submission · Major Revisions

Dear authors,
You are advised to critically respond to all comments point by point when preparing a new version of the manuscript and while preparing for the rebuttal letter. Please address all the comments/suggestions provided by the reviewers.

Kind regards,
PCoelho

Reviewer 1 ·

Basic reporting

no comment

Experimental design

To show the robustness of your approach, the work can be tested on other popular datasets.

Validity of the findings

no comment

Additional comments

Future prospects of the proposed approach should be discussed.

Annotated reviews are not available for download in order to protect the identity of reviewers who chose to remain anonymous.
Cite this review as

Reviewer 2 ·

Basic reporting

While the paper contains valuable research, the overall quality of the manuscript is significantly diminished due to numerous issues related to writing style and grammatical accuracy. Firstly, the writing style is often convoluted, making it challenging for the reader to follow the arguments and understand the key points. Repetitive wording in a sentence (e.g. adversarial) should be revised for clarity and conciseness. Sentences tend to be overly complex, which not only obscures the main ideas but also leads to confusion. A more concise and straightforward style would greatly enhance the clarity and readability of the paper. Moreover, there are frequent grammatical errors throughout the manuscript. These include incorrect verb tenses, subject-verb agreement issues, and improper use of punctuation.
Here are a few examples:
• There are formatting issues, right in the first lines of the paper (abstract), where there are missing spaces after periods and before parentheses.
• For instance, in line 40, incorrect subject-pronoun agreement: "The digital content themselves obtain characteristic information …" should be "The digital content itself obtains characteristic information …"
• The same title is found in lines 401 and 428?
It would be beneficial for the authors to add a paragraph at the end of the introduction section to describe the paper's structure and sections.
Authors need to confirm that all acronyms are defined before being used for the first time.
The paper would benefit from a thorough proofreading and editing process to address these issues. Seeking assistance from a professional editor or utilizing advanced grammar-checking tools could be beneficial. Improving these aspects will help in better conveying the study’s findings and enhancing its impact on the scientific community.

Experimental design

No comment at this point

Validity of the findings

No comment at this point

Cite this review as

Reviewer 3 ·

Basic reporting

• I would suggest authors to include a comparative table of prior works considering different parameters.

• At the end of the introduction, it would be nice to include the structure of the paper.

Experimental design

• More scientific reasoning should be added in the experimental results' explanations.

• Improve your discussion of results through the cause and effect of the modeling parameters, setting values, and model hyper-parameters.

Validity of the findings

• Models’ validation section needs to be enriched with more details and dataset.

Additional comments

• Authors missed many recent references such as in the reference section, e.g.,

Akhtar Z. Deepfakes generation and detection: a short survey. Journal of Imaging. 2023 Jan 13;9(1):18.

Abbas F, Taeihagh A. Unmasking deepfakes: A systematic review of deepfake detection and generation techniques using artificial intelligence. Expert Systems With Applications. 2024 May 18:124260.

Cite this review as

Reviewer 4 ·

Basic reporting

1. The background section could be separated from the current introduction section to improve clarity.
2. The plots for experimental results are difficult to read. The histograms and legends are not presented in a straightforward way.

Experimental design

1. The algorithm would benefit from further discussion on reducing the generation time of adversarial examples.
2. While the DADFI algorithm demonstrates good transferability across different deepfake models, its performance could potentially be enhanced by exploring additional techniques or modifications to the training process.
3. The aggressiveness of the adversarial samples warrants further evaluation. Investigating methods to maintain or improve aggressiveness without requiring the fine-tuning step could be a valuable area for enhancement.

Validity of the findings

1. The limitations and discussion of future work appear to be missing.
2. Using a wider range of deepfake models and datasets would be better to ensure the algorithm's robustness and generalizability.

Additional comments

N/A

Cite this review as

---

## Round 0.2 · accepted · Accept

Dear authors, we are pleased to verify that you meet the reviewer's valuable feedback to improve your research.

Thank you for considering PeerJ Computer Science and submitting your work.

Reviewer 2 ·

Basic reporting

I have checked the revised manuscript and confirm that the authors have implemented all requested changes. The manuscript has been significantly improved.

The authors could improve the visibility of Figures 3, 4 and 5. It is very difficult to distinguish different datasets (CASIA and CelebA) based only on the slightly changed colour of the borders. This problem is particularly pronounced in monochrome printing, which most of us practice. They should change the line type for one data set. Also, Figure 3 could have a more detailed caption of experimental results.

Experimental design

no comment

Validity of the findings

no comment

Cite this review as

Reviewer 3 ·

Basic reporting

The authors have revised the paper to address the comments of reviewers. It appears good now.

Experimental design

Authors have reported and explain the results sufficiently.

Validity of the findings

Authors have reported and explain the results sufficiently.

Additional comments

N/A

Cite this review as

Reviewer 4 ·

Basic reporting

1. Background should be after Introduction.

Experimental design

My previous feedback has been addressed, and everything looks good now.

Validity of the findings

My previous feedback has been addressed, and everything looks good now.

Cite this review as